# Towards Budget-Aware Agentic AI:
# Optimizing Tool Use with Intention-Based World Models

## Abstract

We study budget-constrained tool-augmented agents, where a large language model must solve multi-step tasks by invoking external tools under a strict monetary budget. We formalize this setting as sequential decision making in context space with priced and stochastic tool executions, making direct planning intractable due to massive state–action spaces, high variance of outcomes and prohibitive exploration cost.

To address these challenges, we propose INTENT, an inference-time planning framework that leverages an intention-aware hierarchical world model to anticipate future tool usage, risk-calibrated cost, and guide decisions online. Across cost-augmented StableToolBench, INTENT strictly enforces hard budget feasibility while substantially improving task success over baselines, and remains robust under dynamic market shifts such as tool price changes and varying budgets. To facilitate future research on budget-aware agentic AI, we publicly release our code and datasets: https://anonymous.4open.science/r/icml_agent-A77F.

## 1. Introduction

Large language models are rapidly evolving into agentic systems that can autonomously decompose complex tasks, interact with external tools, and execute multi-step plans. Recent systems demonstrate strong capabilities in deep research (OpenAI, 2025; Team et al., 2025), software engineering (Yang et al., 2024), and web automation (Google, 2026), where reasoning and tool use are tightly interleaved.

Tools constitute the primary interface through which agents interact with the external world and incur real-world im-

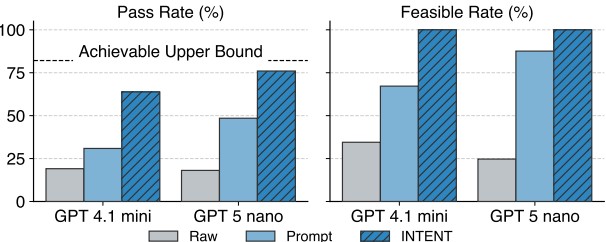

*Figure 1.* Budget awareness of agentic language models on tool cost-augmented StableToolBench. Standalone agents frequently violate hard budget constraints, and prompt-based cost feedback remains insufficient to guarantee budget feasibility or approach the achievable performance upper bound. Our lightweight online planning framework INTENT helps bridge this gap.

pact. With the emergence of standardized protocols such as MCP (Anthropic, 2025) and large-scale tool marketplaces (RapidAPI, 2014; Market, 2025; MCP.so, 2025), agents now have access to thousands of heterogeneous APIs. While this dramatically expands the action space, it also introduces a critical but underexplored dimension: ECONOMIC COST. Unlike token generation, whose marginal cost continues to decline drastically (Appenzeller, 2024; Cottier et al., 2025), many tools expose scarce and monetized resources, such as real-time financial market feeds, high-resolution satellite imagery or irrevocable blockchain state changes.

As agentic systems mature, the central question is no longer whether they can solve complex tasks, but whether we can delegate economically consequential decisions to them. A fundamental question arises: **Can we trust agentic models to make cost-sensitive tool-use decisions on our behalf?**

To probe this question, we instantiate a budgeted tool-use setting on top of the widely used StableToolBench (Qin et al., 2023; Guo et al., 2024), where each task comes with a hard budget constraint and diverse per-call tool prices. As shown in Figure 1, our findings reveal a significant gap: even when explicitly provided with budget feedback after each tool call (PROMPT), strong models frequently exceed the budget due to repetitive retries and unproductive exploration. More advanced reasoning models exhibit better compliance, but only by becoming overly conservative, leaving a large performance gap to the achievable upper bound.

---

[1]Anonymous Institution, Anonymous City, Anonymous Region, Anonymous Country. Correspondence to: Anonymous Author <anon.email@domain.com>.

Preliminary work. Under review by the International Conference on Machine Learning (ICML). Do not distribute.

This setting exposes a non-trivial challenge. Agents must make sequential tool-use decisions under uncertainty, where actions may incur real economic cost, tool outcomes are stochastic, and neither free interaction nor retraining is available at inference time. Moreover, the tool market itself is *dynamic*: available tools and their prices vary with tasks, and new tools may appear without prior experience.

Motivated by these observations, in this work, we study a concrete instantiation of cost-sensitive tool use that captures these challenges while remaining amenable to principled analysis. Each task consists of a user query and a task-specific tool market, where finite available tools have heterogeneous *per-call* prices. The agent must decide *which* tools to invoke, in *what order*, and *when* to terminate, so as to solve the task under a hard budget constraint.

At first glance, this problem appears amenable to several natural solution strategies. One may attempt to cast it as an online knapsack (Buchbinder & Naor, 2009; Agrawal et al., 2009) or linear programming variant, allocating the budget across tools based on their shadow prices. However, such formulations assume independent, additive utilities and fail to capture the strong sequential dependencies between tool calls, where the value of an action is primarily determined by the information it enables for subsequent decisions.

Alternatively, one might hope to endow agents with budget awareness through post-training (Schulman et al., 2017; Shao et al., 2024) or classic constrained reinforcement learning (Altman, 1999; Achiam et al., 2017). Yet this approach is ill-suited to our setting, where tool availability and prices vary across tasks, new tools may appear without prior data, and the overhead of retraining is almost infeasible.

Classical online planning methods such as Monte Carlo Tree Search (Silver & Veness, 2010; Lee et al., 2018) offer a principled way to reason about long-horizon decisions under uncertainty. However, they typically rely on free environment interaction and extensive branching, resulting in prohibitive latency for agentic settings with large action spaces and expensive tool calls.

Taken together, these limitations point to a narrow but crucial design space: a solution must operate purely at inference time, reason about future costs under stochastic tool outcomes, and remain lightweight enough to guide a strong pretrained agent without exhaustive search.

Thus, in this work, we propose INTENT, a lightweight inference-time planning framework for budget-aware tool use. INTENT leverages a learned language world model to simulate tool outcomes and performs calibrated Monte Carlo lookahead to estimate future costs. Crucially, INTENT introduces an intention-based decomposition that separates whether a tool call satisfies the agent's semantic intention from the concrete content of the tool output, enabling accurate cost estimation in highly stochastic environments.

Our contributions are summarized as follows:

- We formalize budget-constrained tool use as a sequential decision problem in agentic language models.

- We propose INTENT, an intention-based planning algorithm that enables budget-aware decision making without retraining or environment interaction.

- We demonstrate substantial performance improvements on StableToolBench across diverse budgets and market settings, approaching the empirical upper bound with mild overhead.

## 2. Model

In this section, We formalize budget-constrained agentic tool use as a sequential decision making over a growing textual history. Each task instance specifies a user query, a hard budget constraint, and a snapshot of a dynamic tool marketplace with per-call costs. The agent follows a interleaved thinking loop that alternates between reasoning, tool calls, and stochastic observations returned by external tools, until it terminates with a final answer. This abstraction captures two core challenges of real-world agentic systems: an unbounded action space induced by free-form arguments, and stochastic transitions arising from tool execution.

### 2.1. Contextual State

We represent the agent's interaction with the environment as sequential decision making over a growing textual context.

**Context Space.** Let $\mathcal{V}$ denote the discrete vocabulary of tokens, and let $\mathcal{V}^*$ denote the set of all finite token sequences. We define the concatenation of two sequences $x_1, x_2 \in \mathcal{V}^*$ as $[x_1, x_2]$, and extend this notation naturally to multiple sequences $[x_1, x_2, \ldots, x_n]$. The agent's state is represented implicitly by its entire interaction history, which grows monotonically through concatenation.

**Serialization Convention.** Throughout this paper, abstract objects that the agent interacts with, such as a tool specification $T$ and execution feedback $o$, are assumed to admit a canonical textual serialization. For simplicity, we identify each object with its serialized token sequence, and use the same symbol to denote both, whenever no ambiguity arises. This convention allows us to embed heterogeneous concepts uniformly into the same context space.

**Language Model.** The agent is powered by a large language model parameterized by $\theta$. We view the LLM as a probabilistic policy operating over the context space. Given a context sequence $h \in \mathcal{V}^*$, the probability of generating a continuation $x \in \mathcal{V}^*$ is denoted by $P_\theta(x \mid h)$.

This formulation enables us to treat reasoning traces, tool calls, and final answers uniformly as token sequences generated by a single policy over the contextual state.

## 2.2. Dynamic Tool Marketplace

We next formalize the environment in which the agent operates, stressing the dynamic availability and pricing of tools.

**Market Snapshot.** We assume a universe of all potential tools $\mathcal{T}$. Upon the arrival of each user query $q$, the agent is presented with a market snapshot, denoted by $\mathcal{M}$. This snapshot specifies the subset of tools that are accessible at that moment (determined by retrieval mechanisms, user permissions, or provider status), together with their current *per-call* costs. Formally, $\mathcal{M} = \{(T^{(j)}, c^{(j)})\}_{j=1}^m$, where each $T^{(j)} \in \mathcal{T}$ encodes the tool's specification (e.g., description, input schema, and usage examples), and $c^{(j)} \in \mathbb{R}_{\geq 0}$ denotes the financial cost incurred by a single invocation of the tool.

**Task Instance.** A task instance is defined as a tuple $\mathcal{I} = (q, B, \mathcal{M})$, sampled from a task distribution $\mathcal{D}$. Here, $q$ denotes the user's natural language query, $B \in \mathbb{R}_{>0}$ is a hard budget constraint, and $\mathcal{M}$ specifies the tool market faced by the agent for this particular request.

## 2.3. Budget-Constrained Agent

Given a task instance $\mathcal{I} = (q, B, \mathcal{M})$, we model the agent's problem-solving process as a ReAct-style (Yao et al., 2022) sequential decision-making procedure over a growing textual history. The interaction unfolds over discrete time steps $t = 1, 2, \ldots$, during which the agent alternates between internal reasoning, external tool use, and observation.

**History Initialization.** Rather than maintaining an abstract state representation, we treat the agent's state as its full interaction history. The initial history $h_0$ consists of the system prompt, the user query, the budget constraint, and the market snapshot: $h_0 = [\texttt{System}, q, B, \mathcal{M}]$.

**Reasoning and Action.** At each step $t$, the agent first generates a reasoning trace $r_t$ to plan or reflect, conditioned on the current history, $r_t \sim \pi_\theta(\cdot \mid h_t)$. Based on both the history and the reasoning trace, the agent then selects a structured action $a_t \sim \pi_\theta(\cdot \mid [h_t, r_t])$. The action space $\mathcal{A}$ consists of two types of operations:

(i) TOOL CALL. $a_t = (\text{CALL}, T_t, u_t)$, where $T_t \in \{T^{(j)}\}_{j=1}^m$ is a tool available in the current market $\mathcal{M}$ and $u_t$ denotes its arguments. (ii) TERMINATION. $a_t = (\text{ANSWER}, y)$, where $y$ is the final response to the user.

**State Transition.** If the agent chooses to terminate, the episode ends. If the agent chooses to call a tool, the transition is governed by an external environment $\mathcal{E}$, which acts as a chance node. The tool execution produces a *stochastic* observation $o_t \sim \mathcal{E}(\cdot \mid T_t, u_t)$, and incurs a cost $\text{COST}(a_t) = c^{(j)}$, where $T_t = T^{(j)}$. The history is then deterministically updated by appending the interaction block: $h_{t+1} = [h_t, r_t, a_t, o_t]$.

## 2.4. Task Formulation

A trajectory $\tau$ is defined as the sequence of interaction components generated until termination at step $K$, $\tau = [h_0, r_1, a_1, o_1, \ldots, a_K]$, where $a_K$ is a ANSWER action.

**Reward.** The agent's objective is to produce a high-quality answer while strictly satisfying the budget constraint. Let $J(a_K, q) \in [0, 1]$ denote an external evaluation metric (e.g., a human evaluator or a judge model) that assesses whether the user query $q$ has been successfully resolved. We define the reward function as

$$R(\tau) = J(a_K, q) \cdot \mathbb{I}\left(\sum_{t=1}^K \text{COST}(a_t) \leq B\right),$$

where $\mathbb{I}(\cdot)$ is an indicator function enforcing a hard budget constraint. Trajectories that exceed the budget receive zero reward regardless of answer quality.

**Optimization Objective.** Our goal is to find a policy $\pi_\theta$ that maximizes the expected reward over the task distribution:

$$\max_\pi \mathbb{E}_{\mathcal{I} \sim \mathcal{D}}\left[\mathbb{E}_{\tau \sim \pi, \mathcal{E}}\left[R(\tau)\right]\right].$$

Directly optimizing this objective in post-training stage is impractical. The action space induced by free-form tool arguments is extremely large, making exploration and credit assignment prohibitively expensive. More importantly, the agent operates in a non-stationary tool market: available tools and their per-call costs vary across task instances, and new tools may appear without prior training data. As a result, a policy optimized offline cannot reliably adapt to the market configuration faced at inference time.

These challenges motivate an online and lightweight inference-time planning approach. Instead of modifying the agent's parameters, we leverage a learned world model to approximate environment transitions and dynamically guide tool selection under budget constraints.

## 3. Methodology

We now introduce an inference-time planning framework for budget-constrained tool use. Our design goal is to minimally intervene on a strong pretrained agent, while dynamically enforcing hard budget constraints under a non-stationary tool market. To this end, we perform lightweight lookahead simulations using a learned world model to anticipate future tool usage and guide decision making online. This section focuses on the conceptual design of the method. Figure 2 outlines and contrasts candidate frameworks, while

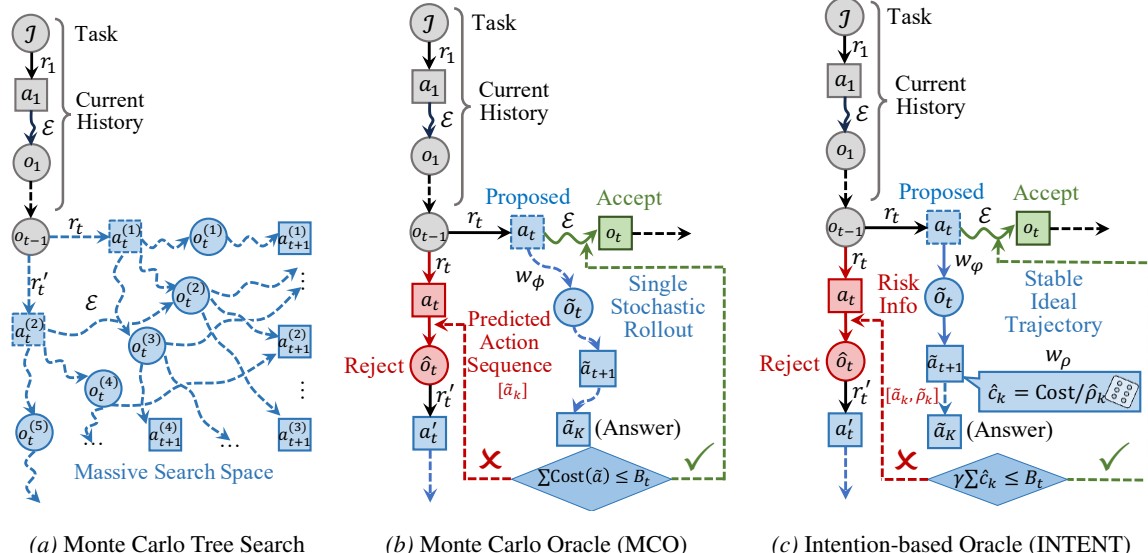

*Figure 2.* Inference-time planning paradigms for budget-aware agentic tool use. (a) MCTS explores a large stochastic search tree with prohibitive cost. (b) MCO enforces budgets via a single stochastic rollout using a language world model. (c) INTENT extracts the agent's latent plan through ideal trajectory simulation and applies intention-aware, risk-adjusted cost estimation for stable budget control.

implementation details, including training and algorithmic pseudocode, are provided in the Appendix A.

### 3.1. Language World Model

First, we train a world model $\mathcal{W}_\phi$, parameterized by $\phi$, to predict the outcome of tool executions. While LLMs may not perfectly simulate the factual accuracy of external tools (e.g., a specific stock price), they excel at predicting the *format* and *structure* of outputs, which is sufficient to elicit the agent's latent planning logic. Given tool call $T_t \in \mathcal{M}$, and arguments $u_t$, the model predicts the observation:

$$\hat{o}_t \sim \mathcal{W}_\phi(\cdot \mid [T_t, u_t]).$$

Importantly, we do not employ the language world model (LWM) for exhaustive tree search. Classical planning algorithms such as Monte Carlo Tree Search (Hao et al., 2023; Zhou et al., 2023) rely on repeated branching and state reuse, which are infeasible in our setting due to the unbounded action space induced by free-form tool call arguments, and the high inference latency of agents. Instead, we restrict the usage of world model to single-trajectory lookahead simulation, which is sufficient for enforcing budget constraints while remaining computationally lightweight.

### 3.2. Monte Carlo Oracle

Our direct feasible application of the LWM is the **Monte Carlo Oracle (MCO)**. The core philosophy is *minimal intervention*: we assume the agent's policy $\pi_\theta$ is inherently capable of solving the task, and the planning algorithm's role is strictly limited to enforcing budget constraints.

**Mechanism.** When the agent proposes an immediate action $a_t$ under current context $[h_t, r_t]$ and remaining budget $B_t$, MCO performs a single *Lookahead Rollout*. Starting from the current state $h_t$, we alternate between the world model $\mathcal{W}_\phi$ and the agent policy $\pi_\theta$ to generate a simulated future trajectory $\tilde{\tau} = (a_t, \tilde{o}_t, \tilde{r}_{t+1}, \tilde{a}_{t+1}, \tilde{o}_{t+1}, \ldots, \tilde{a}_K)$, where $\tilde{a}_K$ is a terminal ANSWER action. We then evaluate the total projected cost $C(\tilde{\tau}) = \sum_{\tilde{a} \in \tilde{\tau}} \text{COST}(\tilde{a})$.

**Decision and Feedback.** We rely on the *Capability Assumption*: if the agent decides to terminate at step $K$, we assume the gathered information is sufficient. The decision logic is purely budget-based: (i) **Accept.** If $C(\tilde{\tau}) \leq B_t$, the action $a_t$ is allowed to be executed in the real environment; (ii) **Reject.** If $C(\tilde{\tau}) > B_t$, the oracle intercepts $a_t$ and prevents its execution. Crucially, to guide the agent's re-planning, we construct a feedback observation $\hat{o}_t = [a_t, \tilde{a}_{t+1}, \ldots, \tilde{a}_K]$ containing the sequence of simulated actions that led to the budget violation. This feedback exposes the future failure to the agent, prompting it to generate a new reasoning trace $r'_t$ and a more informed action $a'_t$ via $\pi_\theta(\cdot \mid [h_t, r_t, a_t, \hat{o}_t])$.

**Limitation.** MCO relies on a single sample estimate. Due to the *stochastic* nature of tools (e.g., a search engine might return irrelevant results, triggering a costly retry loop), the variance of $\text{COST}(\tilde{\tau})$ is high. A single lucky simulation may underestimate the true expected cost, leading to budget overruns in deployment.

### 3.3. Intention-Based Oracle

To mitigate the high variance of single-sample estimation in MCO, we propose **INTENT**, which is motivated by the

observation that an agent's decision to alter its high-level plan is driven less by the specific tool call outcome and more by whether it *satisfies the intention* encoded in reasoning $r_t$.

**Probabilistic Decomposition.** We introduce a binary latent variable $z_t \in \{0, 1\}$, where $z_t = 1$ indicates satisfaction. Although the true tool response $\mathcal{E}(o_t \mid T_t, u_t)$ is independent of the agent's internal reasoning $r_t$, we introduce $r_t$ into our world model to capture the semantic alignment between the action and the expected outcome. By applying the law of total probability, we factorize the generation process as:

$$P_{\mathcal{W}}(o_t \mid r_t, a_t) = \sum_{z_t \in \{0,1\}} \underbrace{P(o_t \mid a_t, z_t)}_{\text{Generation}} \cdot \underbrace{P(z_t \mid r_t, a_t)}_{\text{Intention}}.$$

Note that in the generation term, we omit $r_t$ based on the assumption that once the success status $z_t$ is determined, the specific content of $o_t$ depends primarily on the tool semantics. This factorization leads to two specialized modules:

1. **Intention Predictor.** Estimates the probability $\rho_t$ that observation $o_t$ produced by the proposed tool call $(T_t, u_t)$ will align with the agent's intention revealed in the reasoning, $\tilde{\rho}_t = \mathcal{W}_\rho(z_t = 1 \mid r_t, T_t, u_t)$.

2. **Conditional Generator.** Generates $o_t$ conditioned on the satisfaction status, $\tilde{o}_t \sim \mathcal{W}_\psi(\cdot \mid [T_t, u_t], z_t)$.

**Ideal Trajectory Simulation.** During inference, instead of conducting stochastic sampling which risks traversing costly failure loops, we perform a deterministic simulation of the *ideal trajectory*. Starting from the current step $t$, we construct a trajectory $\tilde{\tau}^*$ by explicitly forcing the conditional generator to satisfy the intention ($z_k = 1$) at every subsequent step $k \geq t$, $\tilde{o}_k \sim \mathcal{W}_\psi(\cdot \mid [T_k, u_k], z_k = 1)$.

This yields a clean trajectory where every tool call works as intended and proceeds towards the solution without deviation, eliciting the *latent plan* that the agent currently holds.

**Geometric Cost Calibration.** With the latent plan $\tilde{\tau}^*$ extracted, we proceed to estimate its expected budget consumption via a *pessimistic estimation* strategy. We observe that an agent adhering to a specific plan will persistently retry or refine arguments until the intention is met ($z_t = 1$), as documented by Xue et al. (2025) and Jin et al. (2026). By modeling the number of trials as a geometric distribution with a constant initial success probability $\tilde{\rho}_k$ (ignoring potential information gain during retries), we derive an upper bound for the expected cost of each step $\tilde{c}_k = \text{COST}(a_k)/\tilde{\rho}_k$.

Finally, to compare this probabilistic estimatation against the hard budget, we introduce a risk preference parameter $\gamma$. The Oracle accepts the proposed action $a_t$ if it is immediately affordable ($\text{COST}(a_t) \leq B_t$) and the risk-adjusted total cost remains within limits, i.e, $\gamma \sum_{\tilde{a}_k \in \tilde{\tau}^*} \tilde{c}_k \leq B_t$.

Here, $\gamma$ serves as a discount factor on our pessimistic estimate, allowing the system to balance between strict safety ($\gamma \geq 1$) and aggressive goal-seeking ($\gamma < 1$).

If rejected, the feedback mechanism follows the MCO protocol but augments the returned trajectory with predicted success probabilities $\hat{o}_t = [a_t, \tilde{\rho}_t, \ldots, \tilde{a}_K]$, which helps the agent identify high-risk bottlenecks for targeted re-planning.

**Simulation Reuse.** To reduce overhead, we cache the future actions $[\tilde{a}_{t+1}, \tilde{a}_{t+2}, \ldots, \tilde{a}_K]$ in ideal trajectory $\tilde{\tau}^*$ upon acceptance. At the subsequent step, if the agent's proposed action $a_{t+1}$ aligns with the cached anticipation ($a_{t+1} \approx \tilde{a}_{t+1}$), we imply plan continuity. Since the remaining trajectory has already satisfied the risk-adjusted budget constraint, we bypass the simulation and grant immediate approval. We also provides other additional mechanisms to boost empirical time efficiency, see discussion in Appendix A.2.

# 4. Experiments

In this section, we empirically evaluate INTENT in budget-constrained and dynamic tool-market settings. Our experiments are designed to answer three important questions: (i) Can agentic models reliably satisfy hard budget constraints while solving tasks? (ii) How effective is INTENT compared to alternative inference-time strategies? (iii) How robust is INTENT to market perturbations such as price changes, new tools, and varying budgets? All experimental implementation details are provided in the Appendix B.

## 4.1. Experimental Setup

**Dataset.** We conduct our experiments on StableToolBench (Guo et al., 2024; 2025), a stable large-scale benchmark for tool learning that evolves from the widely used ToolBench (Qin et al., 2023). ToolBench provides multi-step tool-use tasks where an agent is required to iteratively generate tool calls and incorporate tool observations to complete a given instruction, and covers over 16k real-world tools collected from RapidAPI, spanning 49 diverse categories. StableToolBench further introduces a cache-based API fallback mechanism, ensuring stable and reproducible evaluation.

Since ToolBench does not provide cost information for tools, we augment each instance with synthetic tool prices. For each query $q$, we fix a moderate budget $B = 50$, use the official retriever (Reimers & Gurevych, 2019; Qin et al., 2023) to recall 20 related tools, and then assign each tool $T^{(j)} \in \mathcal{M}$ a per-call cost $c^{(j)}$ independently sampled from a uniform distribution $U(5, 50)$, forming the market snapshot $\mathcal{M}$ for each instance $\mathcal{I} = (q, B, \mathcal{M})$. We evaluate all methods on the 765 test instances in StableToolBench.

**Baselines.** We categorize our baselines into two groups based on whether the budget constraint is explicitly enforced

during inference: *Soft* baselines and *Enforce* baselines.

*Soft* baselines do not enforce the budget constraint, and instead rely on the model's implicit understanding of budget through prompting. This category includes: (i) RAW, where no cost information is provided to the model; and (ii) PROMPT, where the model is explicitly informed of tool costs and current spending via natural language prompts. These baselines evaluate the model's intrinsic ability to reason about budget without external control mechanisms.

*Enforce* baselines explicitly prevent budget violations by introducing external intervention mechanisms when the agent attempts to exceed the budget. We consider three representative methods: (i) DFSDT (Qin et al., 2023), a heuristic depth-first search strategy that prunes branches leading to budget overflow; (ii) BTP (Zheng et al., 2024), which formulates tool selection as a multi-knapsack problem and allocates call quotas for each tool; and (iii) BATS (Liu et al., 2025), which employs a budget tracker to dynamically adjust agent behaviors under different remaining budget levels.

**Evaluation Metrics.** We evaluate all methods from three perspectives: PERFORMANCE, COST-AWARENESS, and EFFICIENCY, in order to comprehensively assess both task-solving ability and budget-sensitive behaviors.

PERFORMANCE. These metrics evaluate the agent's ability to successfully solve tasks under budget constraints. We report: (i) PASS RATE (**PR**), the percentage of tasks successfully solved; (iii) BUDGET-OPTIMAL PASS RATE (**OR**), defined as the ratio between the number of tasks solved by the agent and the total number of tasks that are solvable under the same budget; and (ii) WIN RATE (**WR**), the proportion of tasks where the agent outperforms the reference solution. Following the original StableToolBench, these metrics are evaluated in a LLM-as-a-Judge (Zheng et al., 2023) paradigm, where both the quality of the final answer and the tool call trace are taken into consideration.

COST-AWARENESS. These metrics measure whether the agent exhibits awareness of budget and cost during decision making. We consider: (i) FEASIBLE RATE (**FR**), the proportion of tasks where the agent does not exceed the budget; (ii) AVERAGE COST (**AC**), the average total cost incurred per task; and (iii) AVERAGE PRICE (**AP**), the average per-call price of selected tools, which reflects whether the agent prefers cheaper alternatives when budget is tight.

EFFICIENCY. These metrics assess the computational efficiency of different methods. We adopt: (i) E2E TIME, the end-to-end time for completing all tasks; (ii) LATENCY, the average completion time per task under multi-threaded execution; and (iii) TOKEN CONSUMPTION, the total number of tokens consumed by agents and oracles during inference. We report the relative ratio compared to the RAW method.

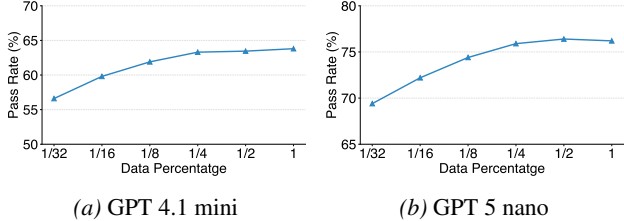

*(a) GPT 4.1 mini*      *(b) GPT 5 nano*

*Figure 3.* Performance under varying amounts of oracle training data, simulating the **introduction of new tools**. Data points correspond to fractions of the full interaction log set (from 1/32 to 1). INTENT shows a clear log-linear scaling trend and strong performance even in the low-data regime, across both backbones.

## 4.2. Main Results

Table 1 summarizes the main results on cost-augmented StableToolBench under both non-reasoning (⑤GPT 4.1 mini) and reasoning (⑤GPT 5 nano) backbones.

**Standalone agents fail to reliably respect budgets.** Table 1 shows that instruction-based methods struggle under hard budget constraints. Although providing explicit cost feedback (PROMPT) improves pass rate over RAW, it still violates budgets in a substantial fraction of tasks (a notable 32.8% for ⑤GPT 4.1 mini) and remains far from the budget-optimal frontier, confirming that implicit budget awareness alone cannot prevent repetitive and unproductive tool use.

**Enforcement introduces performance-efficiency trade-offs.** All *Enforce* baselines achieve perfect feasibility as expected , yet differ markedly in effectiveness and efficiency. Heuristic pruning (DFSDT) and static allocation (BTP) are conservative, leading to limited budget-optimal pass rates. BATS can improve performance but incurs prohibitive inference-time overhead, particularly with reasoning models. These also highlights the limitations of classical online planning in agentic settings.

**INTENT achieves the best overall trade-off.** Across both non-reasoning and reasoning backbones, **INTENT** consistently attains the highest pass rate while strictly respecting budgets. Notably, these gains are achieved with only moderate inference-time overhead, validating intention-level inference-time planning as an effective and practical solution for budget-aware tool use.

## 4.3. Robustness under Dynamic Market

To further evaluate INTENT in realistic and non-stationary environments, we design experiments to study its robustness under three forms of market dynamics: the emergence of new tools, relative price changes, and varying budget levels.

**New tools.** A key motivation for online planning is that real-world tool markets are continually evolving, with new tools appearing that are not covered by the agent's prior

*Table 1.* Main results on the cost-augmented StableToolBench (Guo et al., 2024). We compare our proposed methods (MCO and INTENT) against *Soft* (instruction-based) and *Enforce* (external constraint-based) baselines across both Non-Reasoning (⑤GPT 4.1 mini) and Reasoning (⑤GPT 5 nano) backbones. Comprehensive evaluation covers three dimensions: PERFORMANCE (Pass Rate, Budget-Optimal Pass Rate, Win Rate), COST-AWARENESS (Feasible Rate, Average Cost, Average Price), and EFFICIENCY (E2E Time, Average Latency and Total Token Consumption relative to the RAW baseline). **INTENT** consistently achieves the best performance with budget adherence.

| **Method** | | PERFORMANCE | | | COST-AWARENESS | | | EFFICIENCY | | |
|---|---|---|---|---|---|---|---|---|---|---|
| | | PR↑ | OR↑ | WR↑ | FR↑ | AC↓ | AP↓ | Time↓ | Lat.↓ | Tok.↓ |
| **Non-Reasoning Model** | | | | | | | | | | |
| *Soft* | Raw (Yao et al., 2022) | 19.1 | 23.3 | 37.5 | 34.5 | 102.1 | 28.2 | 1.00× | 1.00× | 1.00× |
| | Prompt | 30.9 | 37.7 | 41.4 | 67.2 | 43.4 | 24.2 | 0.65× | 0.55× | 0.56× |
| *Enforce* | DFSDT (Qin et al., 2023) | 45.9 | 55.9 | 55.4 | 100.0 | 35.2 | 20.2 | 0.97× | 0.99× | 1.14× |
| | BTP (Zheng et al., 2024) | 46.4 | 56.8 | 59.0 | 100.0 | 32.5 | 19.4 | 1.01× | 0.76× | 0.84× |
| | BATS (Liu et al., 2025) | 53.0 | 64.6 | 68.1 | 100.0 | 35.6 | 20.8 | 1.96× | 3.55× | 4.13× |
| | MCO (Ours) | 58.9 | 71.8 | 72.5 | 100.0 | 27.1 | 18.7 | 1.90× | 2.05× | 2.15× |
| | **INTENT (Ours)** | **63.8** | **77.8** | **73.3** | **100.0** | **24.9** | **19.1** | **1.23×** | **1.76×** | **1.70×** |
| **Reasoning Model** | | | | | | | | | | |
| *Soft* | Raw (Yao et al., 2022) | 18.1 | 22.1 | 44.7 | 24.7 | 145.8 | 27.8 | 1.00× | 1.00× | 1.00× |
| | Prompt | 48.5 | 59.1 | 62.0 | 87.6 | 38.1 | 20.1 | 0.64× | 0.60× | 0.76× |
| *Enforce* | DFSDT (Qin et al., 2023) | 57.2 | 69.7 | 71.9 | 100.0 | 35.6 | 18.6 | 1.10× | 0.91× | 1.08× |
| | BTP (Zheng et al., 2024) | 57.7 | 70.3 | 73.2 | 100.0 | 36.2 | 18.8 | 1.22× | 1.67× | 0.87× |
| | BATS (Liu et al., 2025) | 52.8 | 64.3 | 71.7 | 100.0 | 32.6 | 17.3 | 7.67× | 10.1× | 5.76× |
| | MCO (Ours) | 72.0 | 87.7 | 82.3 | 100.0 | 31.9 | 18.1 | 1.87× | 2.23× | 2.28× |
| | **INTENT (Ours)** | **76.0** | **92.6** | **86.1** | **100.0** | **29.2** | **17.9** | **1.79×** | **2.16×** | **2.40×** |

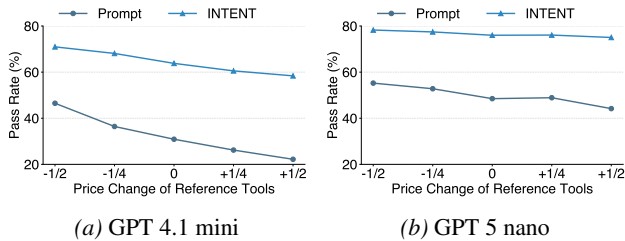

*(a) GPT 4.1 mini*    *(b) GPT 5 nano*

*Figure 4.* Robustness to **relative price changes** of reference tools. We uniformly increase or decrease the prices of reference tools by fixed ratios (from a 50% discount to a 50% markup), while keeping other tools unchanged. INTENT is substantially less sensitive to price perturbations than Prompt across both backbones.

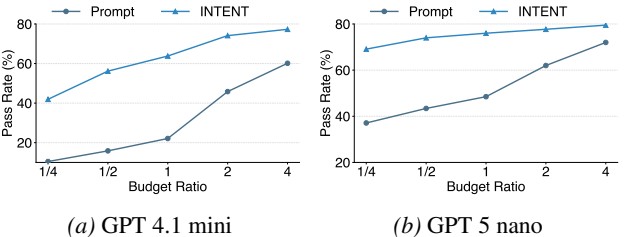

*(a) GPT 4.1 mini*    *(b) GPT 5 nano*

*Figure 5.* Performance under **varying budget levels**. Budgets are scaled by fixed ratios relative to the default setting. INTENT scales effectively with increased budget and achieves competitive performance under tight budgets, across both backbones.

knowledge. Relying solely on parametric knowledge can therefore introduce substantial bias. To simulate this process, we vary the number of interaction logs used to train the proposed oracle, representing different stages of market exposure. Results in Figure 3 show a log-linear scaling trend in the early stage as more data is accumulated. Notably, INTENT already achieves strong performance with only a few thousand logs, indicating that it can rapidly infer tool characteristics from limited interaction data.

**Price adjustments.** In dynamic markets, tools with similar functionality often undergo relative price changes, requir-

ing agents to reason about cost-effectiveness and substitute tools when necessary. We simulate this setting by identifying reference tools annotated in ToolBench and selectively increasing or decreasing their prices, while keeping other retrieved tools unchanged. As shown in Figure 4, PROMPT is highly sensitive to these price perturbations, reflecting path dependence and insufficient exploration in standalone agents. In contrast, INTENT maintains consistently high pass rates with significantly smaller performance degradation, demonstrating robust cost-aware decision making.

**Budget scaling.** Finally, we vary the available budget to reflect different user preferences for the same task. It can

be seen from Figure 5 that, although INTENT is designed for budget-constrained scenarios, it does scale when more resources are available. Moreover, its efficient budget utilization allows INTENT under tight budgets to achieve performance comparable to PROMPT under substantially larger budgets, highlighting its ability to adapt across a wide range of resource regimes.

Overall, these results show that INTENT generalizes robustly across multiple forms of market non-stationarity, while requiring only periodic updates to a lightweight oracle model, without modifying the parameters of the underlying gigantic agentic language model.

## 5. Related Work

**Agentic AI.** Agentic AI typically refers to fronteer LLMs equipped with explicit reasoning and tool-use capabilities to autonomously solve complex, multi-step tasks. Recent systems have demonstrated strong performance in deep research and information synthesis (Li et al., 2025a; Team et al., 2025), GUI control (Qin et al., 2025), and software engineering (Yang et al., 2024; Tao et al., 2024), etc.

On the infrastructure side, standardized protocols such as Model Context Protocol (MCP; Protocol, 2025) have enabled scalable integration of heterogeneous tools, while specialized reinforcement learning frameworks provide environments for developing and evaluating tool-augmented agents (Chai et al., 2025; Jiang et al., 2025; Fu et al., 2025).

Most existing works focus on expanding the capability frontier of agents, e.g., learning to invoke a large variety of tools (Tang et al., 2023; Qin et al., 2023), generating long-horizon tool-use trajectories (Chen et al., 2025; Gao et al., 2025), or coordinating multiple agents for collaborative problem solving (Li et al., 2025b). These methods optimize task success without constraints, and typically allow repeated tool calls until sufficient information is obtained.

In contrast, far less attention has been paid to the *economic dimension* of agentic behavior. While a growing body of work studies the efficiency of agentic systems, including token efficiency via reasoning compression or speculative decoding (Xia et al., 2025; Zhang et al., 2025; Chen et al., 2023; Hu et al., 2025), and tool efficiency through reducing or approximating tool calls (Xu et al., 2025; Nichols et al., 2025)—these approaches mainly optimize computational cost or latency. They do not model explicit tool prices, nor do they reason about *hard budget feasibility*.

As a result, existing agents cannot capture realistic scenarios in which tools are monetized, retries incur irreversible monetary costs, and agents must trade off information gain against expenditure in a dynamic tool market. In this work, we explicitly formalize budget-constrained tool use as a sequential decision problem with hard monetary constraints, and study how a general-purpose agent can operate rationally under such conditions.

**Language World Models.** Language world models (LWMs) are trained to simulate environment dynamics in context space, enabling agents to reason about future outcomes without interacting with the real environment. In training, LWMs have been used to replace expensive or unstable external tools, significantly reducing data collection costs (Guo et al., 2025; Sun et al., 2025). Examples include simulated search engines (Fan et al., 2025; Zhang et al., 2026), synthesized compiler feedback (Pan et al., 2024; Cheng et al., 2026), and general tool environments (Ren et al., 2025; Fang et al., 2025; Xi et al., 2025).

At inference time, LWMs are mainly applied to deterministic environments such as text-based games (e.g., ALFWorld, Shridhar et al., 2020; GridWorld, Sasso et al., 2025), where they support planning via classical algorithms such as MCTS (Dainese et al., 2024; Hao et al., 2023).

Our setting is substantially more challenging: tools exhibit high stochasticity, and repeated failures often trigger costly retry loops. Instead of predicting exact future states, we introduce an intention-based world model that abstracts tool outcomes at the semantic level, focusing on whether a tool call satisfies the agent's intention. This enables reliable cost estimation and budget-aware planning in highly uncertain environments.

## 6. Conclusions

In this work, we highlighted budget-aware tool use as a fundamental yet underexplored problem in agentic AI model design. As agents increasingly rely on external tools in open and dynamic markets to make real-world impact, their ability to reason under hard resource constraints becomes essential for reliable and deployable systems.

To this end, we proposed INTENT, an intention-based inference-time planning framework that views budget control as a problem of anticipating whether future tool interactions will satisfy the agent's high-level intent, rather than predicting exact tool outcomes. This abstraction allows effective budget enforcement with minimal intervention, without retraining or heavy search.

More importantly, our study suggests that budget awareness should be treated as a first-class objective in agentic AI, and that intention-level reasoning provides a natural interface between stochastic environments and resource-constrained decision making. We hope this work draws attention to budget-constrained agentic planning, and encourages further exploration of lightweight, inference-time control mechanisms for real-world agentic systems.

## Impact Statement

This paper presents work whose goal is to advance the budget-awareness of tool-using agents. There might be some potential societal consequences of our work, none which we feel must be specifically highlighted here.

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

# A. Oracle Implementations

## A.1. Training

**Trajectory Collection.** All oracle models are trained on agent interaction logs naturally collected from user conversations. Each log corresponds to a task instance $\mathcal{I} = (q, B, \mathcal{M})$ sampled from the task distribution $\mathcal{D}$, and contains a ReAct-style trajectory $\tau = [h_0, r_1, a_1, o_1, r_2, a_2, o_2, \ldots, a_K]$, where $r_t$ is the reasoning trace, $a_t$ is the agent action, and $o_t$ is the environment observation. No additional data collection or synthetic trajectories are introduced.

**Language World Model.** $\mathcal{W}_\phi$ is instantiated as a large language model fine-tuned on tool interaction records. Each training example consists of a tool call and its observed response $(a_t, o_t)$, and the model is trained by standard next-token prediction to approximate $\mathcal{W}_\phi(o_t \mid [T_t, u_t])$.

**Latent Intention Annotation.** The intention variable $z_t$ is not directly observable. For each triple $(r_t, a_t, o_t)$, we apply an LLM-as-a-Judge (Zheng et al., 2023) to determine whether the observation semantically satisfies the intention of the tool call expressed in $r_t$, yielding a binary label $z_t \in \{0, 1\}$.

**Conditional Generator.** $\mathcal{W}_\psi$ is implemented as a large language model trained on positive intention samples $(a_t, o_t)$ with $z_t = 1$, learning the conditional distribution $\mathcal{W}_\psi(o_t \mid [T_t, u_t], z_t = 1)$.

**Intention Predictor.** $\mathcal{W}_\rho$ is an encoder-only Transformer with a classification head. It is trained as a binary classifier on $(r_t, a_t, z_t)$ using standard cross-entropy loss to estimate the success probability $\tilde{\rho}_t = P(z_t = 1 \mid [r_t, T_t, u_t])$. Moreover, to obtain well-calibrated probabilities, we apply post-hoc temperature scaling (Guo et al., 2017) on a held-out validation set. The calibrated score $\tilde{\rho}_t$ is used for geometric cost estimation in the oracle.

## A.2. Additional Mechanisms

In addition to the core oracle logic, we adopt several lightweight mechanisms in implementation to reduce redundant rollouts and unnecessary exploration. These mechanisms do not alter the underlying decision rules of the oracle, but improve computational efficiency in practical deployments. We incorporate these mechanisms in the main algorithm.

**Rollout Cache.** As described in the Section 3.3, we cache the future action sequence $\mathcal{C}_t = [\tilde{a}_{t+1}, \tilde{a}_{t+2}, \ldots, \tilde{a}_K]$ obtained from the ideal trajectory $\tilde{\tau}^*$ upon acceptance. If the agent's subsequent proposal satisfies $a_{t+1} \approx \tilde{a}_{t+1}$ (e.g., tolerant for some argument mismatches), we imply plan continuity and skip simulation, directly approving the action (as long as the action is budget feasible, i.e.,

$\text{COST}(a_{t+1}) \leq B_t)$, and then the cache queue pops to $\mathcal{C}_{t+1} \leftarrow [\tilde{a}_{t+2}, \ldots, \tilde{a}_K]$, accordingly. If a cache miss happens instead, the rollout cache is emptied $\mathcal{C}_{t+1} \leftarrow []$, and will be replace by the new rollout $\mathcal{C}_{t+1} \leftarrow [\tilde{a}'_{t+2}, \ldots, \tilde{a}'_K]$ if the current action $a_{t+1}$ gets approved.

**Last Call Cache.** Let $\bar{a}_t$ denote the last rejected action recorded at step $t$ (or $\varnothing$ if none). Upon a rejection at step $t$, we set $\bar{a}_{t+1} \leftarrow a_t$. If at step $t + 1$ the agent proposes the same action again $a_{t+1} = \bar{a}_{t+1}$, this might be interpreted as a strong evidence that the action is necessary under current situation in the agent's view. Therefore we directly accept it (provided $\text{COST}(a_{t+1}) \leq B_{t+1}$) without additional rollout, and then clear the cache $\bar{a}_{t+2} \leftarrow \varnothing$.

**Blacklist.** During rollout, for any action $a_t = (\text{CALL}, T_t, u_t)$ with predicted success probability $\tilde{\rho}_t < \delta$, we add the corresponding tool to a blacklist set $\mathcal{B} = \mathcal{B} \cup \{T_t\}$. All tools in $\mathcal{B}$ are permanently excluded from the candidate action space for the current task instance $\mathcal{I}$, i.e., $\mathcal{M} \leftarrow \mathcal{M} \setminus \mathcal{B}$.

## A.3. Algorithms

In this section, we provide the detailed pseudocode for the proposed framework and the oracle implementations.

**Budgeted-Constrained Agent.** Algorithm 1 summarizes the overall execution loop of an oracle-guided, budget-constrained agent. At each step, the agent follows a standard ReAct-style (Yao et al., 2022) interaction pattern and proposes an action, which is then intercepted by an oracle for budget feasibility checking. The oracle operates purely at inference time and does not modify the agent policy, but only decides whether to ACCEPT the costly tool call to be executed in the real environment or return a synthetic feedback signal for re-planning. Algorithm 2 and Algorithm 3 instantiate the two concrete oracle implementations introduced in the Section 3, namely the Monte Carlo Oracle (MCO) and the proposed Intention-Based Oracle (INTENT).

Notably, in Algorithm 3, we explicitly distinguish between the core components and the engineering optimizations. The core logic is highlighted in **blue**. The auxiliary mechanisms introduced to improve efficiency, i,e. the *Rollout Cache*, *Last Call Cache*, and *Blacklist*, are marked in **gray**.

# B. Implementation Details

## B.1. Training Details

**Data.** We train all oracle components using interaction logs provided by StableToolBench (Guo et al., 2024). Specifically, for the *Language World Model* and the *Conditional Generator*, we use the MirrorAPI-Cache training split (Guo et al., 2025), which consists of approximately 100k real tool interaction trajectories collected from RapidAPI. Following

**Algorithm 1** Oracle-Guided Budget-Constrained Agent
___
**Require:** Task instance $\mathcal{I} = (q, B, \mathcal{M})$, Agent policy $\pi_\theta$, Environment $\mathcal{E}$

1: **Initialize:** History $h_0 \leftarrow [\texttt{System}, q, B, \mathcal{M}]$, Step $t \leftarrow 1$, Current budget $B_t \leftarrow B$
2: **while** $B_t \geq 0$ **do**
3:     **Reasoning:** $r_t \sim \pi_\theta(\cdot \mid h_t)$
4:     **Action:** $a_t \sim \pi_\theta(\cdot \mid [h_t, r_t])$
5:     **if** $a_t$ is (ANSWER, $y$) **then**
6:         **return** $y$   ▷ Terminate and return final answer
7:     **end if**
8:     // CONSULT THE SPECIFIC ORACLE IMPLEMENTATION (MCO OR INTENT)
9:     decision, $\hat{o}_t \leftarrow \text{ORACLE}(h_t, r_t, a_t, B_t)$
10:     **if** decision = ACCEPT **then**
11:         **Execute:** $o_t \sim \mathcal{E}(\cdot \mid a_t)$  ▷ Real tool execution
12:         $B_{t+1} \leftarrow B_t - \text{COST}(a_t)$
13:     **else**
14:         **Intervention:** $o_t \leftarrow \hat{o}_t$    ▷ Oracle returns simulated failure trace
15:         // AGENT WILL RE-PLAN BASED ON THIS FEEDBACK IN NEXT ITER
16:     **end if**
17:     **Update:** $h_{t+1} \leftarrow [h_t, r_t, a_t, o_t]$
18:     $t \leftarrow t + 1$
19: **end while**
20: **return** FAILURE          ▷ Budget exhausted

**Algorithm 2** Monte Carlo Oracle (MCO)
___
1: **function** ORACLE($h_t, r_t, a_t, B_t$)
2:     **Lookahead:** Simulate trajectory $\tilde{\tau}$ starting from $h_t$
3:     $\tilde{\tau} \leftarrow [a_t]$
4:     **while** $a \leftarrow \text{LAST}(\tilde{\tau})$ is not ANSWER **do**
5:         $\tilde{o} \sim \mathcal{W}_\phi(\cdot \mid a)$    ▷ World Model Prediction
6:         $\tilde{r}, \tilde{a} \sim \pi_\theta(\cdot \mid [h_t, r_t, \tilde{\tau}, \tilde{o}])$    ▷ Agent Policy
7:         $\tilde{\tau} \leftarrow [\tilde{\tau}, \tilde{o}, \tilde{r}, \tilde{a}]$
8:     **end while**
9:     **Cost Estimation:** $\text{COST}(\tilde{\tau}) \leftarrow \sum_{a \in \tilde{\tau}} \text{COST}(a)$
10:     **if** $\text{COST}(\tilde{\tau}) \leq B_t$ **then**
11:         **return** ACCEPT, $\varnothing$
12:     **else**
13:         $\hat{o}_t \leftarrow \text{EXTRACTACTIONS}(\tilde{\tau})$    ▷ Return predicted future action sequence as hints
14:         **return** REJECT, $\hat{o}_t$
15:     **end if**
16: **end function**

**Hardware.** All training and inference experiments are conducted on a single NVIDIA RTX Pro 6000 GPU.

### B.2. Algorithm Implementation

**DFSDT.** The original DFSDT implementation does not explicitly account for monetary budgets. We augment it with a budget enforcement mechanism: if a proposed action incurs a cost exceeding the remaining budget, the corresponding branch is immediately pruned. The search width is fixed to 10 in all experiments.

**BTP.** We implement BTP on top of the DFSDT framework. The experience memory is constructed from the same Reproduction Data used in our experiments. Tool scores are computed following the original formulation, using GPT-4.1-mini as the evaluator. Tool similarity is measured using Qwen3-0.6B-Embedding. We enable the Blacklist mechanism with threshold $\tau = 0.15$, consistent with the original setting.

**BATS.** We faithfully reproduce the original BATS pipeline and prompts. Since BATS assumes per-tool budgets, we adapt it to a unified global budget constraint. To control inference overhead, we cap the number of Verification Agent calls at $K = 5$ per iteration.

**MCO.** For Monte Carlo Oracle, the world model sampling temperature is set to 1.0. We enable the Rollout Cache to reuse simulated trajectories across steps.

**INTENT.** For INTENT, the Conditional Generator sampling temperature is set to 0.3. We enable both the Rollout Cache and the Blacklist mechanism. Across all experiments, we fix the risk discount factor $\gamma = 0.5$ and the rejection tolerance $\delta = 0.1$. No task-specific or dataset-specific hyperparameter

prior work, we employ GPT-4.1-mini as an LLM-as-a-judge to annotate intention satisfaction and retain 28k trajectories where the tool calls are deemed successful.

For the *Intention Predictor*, we construct a separate dataset from the ToolBench Reproduction Data (Qin et al., 2023), yielding 86k (Thought, Action, Observation) triples. Each triple is annotated using GPT-4.1-mini to determine whether the tool outcome satisfies the agent's expressed intention. All datasets are split into training, validation, and test sets with an 8:1:1 ratio. The validation split is used for checkpoint selection and post-hoc probability calibration.

**Model Architecture.** Both the Language World Model and the Conditional Generator are instantiated using **Qwen2.5-3B-Instruct**. The Intention Predictor is implemented as an encoder-only model based on **Qwen3-0.6B-Embedding**.

**Optimization.** For the Language World Model and Conditional Generator, we use a batch size of 64 and a learning rate of $7 \times 10^{-5}$, with a linear warmup over the first 5% of steps followed by cosine decay. Both models are trained for two epochs with full-parameter fine-tuning. The Intention Predictor is trained with batch size 32, learning rate $5 \times 10^{-5}$, and two epochs. All models are full parameter fine-tuned.

tuning is performed.

## B.3. Evaluation Details

**Automatic Evaluation.** Pass Rate and Win Rate are evaluated using the same logic as SoPR and SoWR in Stable-ToolBench, with two modifications: (i) Solutions are additionally required to satisfy the budget constraint, and (ii) The evaluator model is upgraded to GPT-4.1-mini. Prior work has shown strong agreement between this evaluation protocol and human judgments (Qin et al., 2023). For Win Rate, the reference solution is generated by the PROMPT baseline using GPT-5-mini. All evaluations are conducted using Major Voting @ 3 aggregation.

**Budget-Optimal Pass Rate.** To estimate the Achievable Upper Bound used in the Budget-Optimal Pass Rate metric, we perform an exhaustive search using DFSDT with width $w = 10$, powered by GPT-5-nano. For each query, we enumerate tool-use trajectories until collecting five unique solutions that satisfy the budget constraint. If at least one solution is judged correct by the evaluator, the query is marked as solvable under the given budget. This procedure yields an empirical upper bound on achievable performance.

---

**Algorithm 3** Intention-Based Oracle (**INTENT**)

1: **Global:** Risk Factor $\gamma$, Rollout Cache $\mathcal{C} \leftarrow []$, Last Rejected $\bar{a} \leftarrow \varnothing$, Blacklist $\mathcal{B} \leftarrow \emptyset$, Blacklist Threshold $\delta$, Market Snapshot $\mathcal{M}$
2: **function** ORACLE($h_t, r_t, a_t, B_t$)
3:     // LAST CALL CACHE
4:     **if** $a_t \approx \bar{a}$ **and** COST($a_t$) $\leq B_t$ **then**
5:         **Global** $\bar{a} \leftarrow \varnothing$; **return** ACCEPT, $\varnothing$
6:     **end if**
7:     // ROLLOUT CACHE (SIMULATION REUSE)
8:     **if** $\mathcal{C}$ is not empty **and** $a_t \approx \mathcal{C}[0]$ **then**
9:         Pop $\mathcal{C}[0]$
10:         **return** ACCEPT, $\varnothing$       ▷ Plan continues
11:     **else**
12:         $\mathcal{C} \leftarrow []$       ▷ Cache miss, clear cache
13:     **end if**
14:     **Ideal Trajectory Simulation:**
15:     Initialize $\tilde{\tau}^* \leftarrow []$, $k \leftarrow t$, $a_k \leftarrow a_t$, $r_k \leftarrow r_t$
16:     Total Expected Cost $\sigma \leftarrow 0$
17:     **loop**
18:         // 1. INTENTION PREDICTION
19:         $\tilde{\rho}_k \leftarrow \mathcal{W}_\rho(z = 1 \mid [r_k, T_k, u_k])$
20:         // BLACKLIST UPDATE
21:         **if** $\tilde{\rho}_k < \delta$ **then**
22:             $\mathcal{B} \leftarrow \mathcal{B} \cup \{T_k\}$
23:             $\mathcal{M} \leftarrow \mathcal{M} \setminus \mathcal{B}$
24:         **end if**
25:         // 2. GEOMETRIC COST CALIBRATION
26:         $\tilde{c}_k \leftarrow$ COST($a_k$)$/\tilde{\rho}_k$
27:         $\sigma \leftarrow \sigma + \tilde{c}_k$
28:         // 3. CONDITIONAL GENERATION
29:         $\tilde{o}_k \sim \mathcal{W}_\psi(\cdot \mid [T_k, u_k], z = 1)$
30:         $\tilde{\tau}^* \leftarrow [\tilde{\tau}^*, r_k, a_k, \tilde{o}_k]$
31:         **Agent Step:**
32:         $\tilde{r}_{k+1}, \tilde{a}_{k+1} \sim \pi_\theta(\cdot \mid [h_t, \tilde{\tau}^*], \mathcal{M})$
33:         **if** $\tilde{a}_{k+1}$ is ANSWER **then**
34:             **break**
35:         **end if**
36:         $k \leftarrow k + 1$, $a_k \leftarrow \tilde{a}_{k+1}$, $r_k \leftarrow \tilde{r}_{k+1}$
37:     **end loop**
38:     **Decision:**
39:     **if** COST($a_t$) $\leq B_t$ **and** $\gamma \cdot \sigma \leq B_t$ **then**
40:         $\mathcal{C} \leftarrow [\tilde{a}_{t+1}, \ldots, \tilde{a}_K]$   ▷ Update Rollout Cache
41:         **Global** $\bar{a} \leftarrow \varnothing$
42:         **return** ACCEPT, $\varnothing$
43:     **else**
44:         $\hat{o}_t \leftarrow [a_t, \tilde{\rho}_t, \ldots, \tilde{a}_K]$   ▷ Failure risk feedback
45:         **Global** $\bar{a} \leftarrow a_t$         ▷ Record rejection
46:         **return** REJECT, $\hat{o}_t$
47:     **end if**
48: **end function**

