# OpenReview forum: "Towards Budget-Aware Agentic AI: Optimizing Tool Use with Intention-Based World Models"
_ICML.cc/2026/Conference — Submitted to ICML 2026_

### Official Review · Reviewer_Wy1B · 2026-02-21

**Soundness:** 3
**Presentation:** 3
**Significance:** 3
**Originality:** 3
**Overall Recommendation:** 4
**Confidence:** 3

**Summary:**

The paper studies budget-aware agentic tool use, where each tool call has a monetary cost and stochastic outcome, and the agent must solve multi-step tasks under a strict total budget. To address this, the authors formalize the problem as sequential decision-making over a context-based state space with a hard budget feasibility requirement. They propose INTENT, an inference-time planning framework that combines Monte Carlo lookahead with an intention-based world model: it predicts whether a tool call will satisfy the current sub-intention, uses this intention abstraction to estimate (and risk-adjust) expected future cost, and selects tool actions that are both effective and budget-feasible. Experiments on StableToolBench with synthetic pricing and fixed budgets show that INTENT improves task success/quality while maintaining perfect budget feasibility, and it is more robust to price variations than prompting-only baselines.

**Compliance With Llm Reviewing Policy:**

Affirmed.

**Final Justification:**

This paper addresses an important and practically relevant problem—budget-aware agentic tool use under strict monetary constraints—and proposes a technically coherent inference-time planning framework based on intention-level cost-aware reasoning. I found the paper’s main strengths to be the relevance of the problem, the reasonable and modular design of the proposed method, and the solid empirical results within the benchmark setting presented in the original submission. At the same time, my concerns centered on the external validity of the pricing setup, the support for some modeling assumptions, the reliability of the evaluation protocol, and the clarity of the paper’s positioning between feasibility guarantees and performance/cost efficiency. While these issues limit the breadth of the paper’s impact and prevent me from rating it higher, they do not outweigh its core merits in my assessment. The authors’ rebuttal was helpful in clarifying several of these points and reinforced my prior understanding of the paper, but it did not fundamentally change my overall evaluation. In light of the discussion about post-submission evidence, my final recommendation is based primarily on the original submission, and I therefore maintain my overall recommendation at 4 (Weak Accept) while placing my confidence at 3. Overall, I continue to view this work as a technically solid and meaningful contribution that is likely to be useful to the community.

**Key Questions For Authors:**

1. **Pricing realism / external validity:** The main experiments use synthetic i.i.d. uniform tool costs under a fixed budget regime. How does INTENT perform under more realistic pricing structures (e.g., long-tailed/log-normal costs, correlated prices within tool categories, or price–quality correlation)?
   *Why this matters:* If INTENT remains clearly better under these settings, my confidence in the broad applicability and significance of the work would increase; if performance degrades substantially, I would view the contribution as more benchmark-specific.

2. **Calibration of intention success probabilities:** Are the intention predictor’s probabilities calibrated (e.g., reliability plots/ECE), and how stable are they across tasks/markets? Do you observe distribution shift effects when tools/prices change, and how does periodic oracle updating mitigate this?
   *Why this matters:* Strong calibration/stability evidence would strengthen the soundness of the cost estimation and risk control; poor calibration would suggest the method’s gains may be fragile and could require additional mechanisms.

3. **Hyperparameter sensitivity (risk control):** The method fixes risk-related hyperparameters (e.g., γ and δ). Can you provide a sensitivity analysis showing the trade-off between success/quality, feasibility, and overhead across different γ/δ values and budget levels?
   *Why this matters:* If INTENT is robust to these choices, I would increase my confidence in deployability and default settings; if it requires careful tuning per scenario, I would downgrade practical significance.

4. **Evaluation reliability (LLM-as-a-judge):** Since Pass/Win metrics rely on an evaluator model and a reference-policy generator, can you provide evidence of evaluation robustness (e.g., multi-judge agreement, small-scale human verification, or consistency across different evaluators)?
   *Why this matters:* Strong robustness checks would increase confidence that reported gains reflect real quality improvements rather than evaluator bias; weak agreement would weaken the empirical support.

5. **Failure modes and attribution:** For cases where INTENT fails (or underperforms), what are the dominant causes (world model prediction error, intention misclassification, tool unreliability, insufficient budget allocation, etc.)?
   *Why this matters:* A clear failure taxonomy and targeted ablations would help judge whether limitations are fundamental (reducing significance) or mostly engineering/estimation issues that can be improved (supporting a more positive overall evaluation).

**Limitations:**

Partially. The paper identifies the general importance of budget constraints, but the limitations discussion could be more explicit and concrete. I recommend strengthening it in three ways:
1) **External validity of the pricing setup:** Clearly state that the main results are obtained under synthetic pricing (e.g., i.i.d. uniform costs and fixed budgets) and discuss how real markets may exhibit long-tailed, correlated, and quality-linked pricing that could affect performance.
2) **Evaluation caveats:** Acknowledge the reliance on LLM-as-a-judge (and reference-policy generation) and add a brief robustness plan (e.g., multi-judge agreement, small human spot-checks).
3) **Potential negative societal impacts / misuse:** Discuss risks such as cost-minimizing behaviors that might encourage low-quality or biased information retrieval, strategic “gaming” of pricing, or reinforcing inequities when access is gated by budget. Also note operational risks (e.g., agents choosing cheaper but less reliable tools) and possible mitigations (auditing, safety/quality constraints, transparency, and monitoring).

**Strengths And Weaknesses:**

## Strengths
1. **Problem relevance and significance:** The paper tackles budget-aware agentic tool use, a practical constraint in real deployments where tool calls have monetary costs and outcomes are stochastic.
2. **Technically reasonable approach:** INTENT introduces an inference-time planning framework that uses an intention-level abstraction to stabilize cost/benefit estimation under stochastic tool outputs, which is a sensible modeling choice for planning under uncertainty.
3. **Strong empirical performance under the given setup:** Across two backbones, INTENT achieves higher task success/quality metrics while maintaining perfect budget feasibility, and does so with lower overhead than heavier enforcement baselines.
4. **Robustness analyses included:** The paper studies sensitivity to tool price perturbations and discusses adaptation to new tools/market updates, which strengthens the practical story beyond a single fixed setting.
5. **Clear high-level narrative and implementability:** The paper motivates why naive prompting can overspend, provides algorithmic descriptions, and presents the method as a modular add-on without finetuning the underlying LLM, making the contribution easier to adopt.

## Weaknesses
1. **Limited external validity of pricing/budget simulation:** The main experiments rely on synthetic pricing (e.g., i.i.d. uniform costs) and a fixed budget regime, which may not reflect realistic long-tail, correlated, or quality-linked pricing structures.
2. **Key assumptions behind cost estimation need stronger validation:** The intention-based success modeling and risk adjustment are plausible, but the paper would benefit from deeper evidence that the estimated success probabilities/costs are well calibrated and stable across tasks and markets.
3. **Evaluation relies heavily on LLM-as-a-judge:** Pass/win metrics depend on an evaluator model and a reference-policy generator, which may introduce bias; more human checks or multi-judge agreement analysis would strengthen the claims.
4. **Hyperparameter sensitivity is under-explored:** The method fixes risk-related thresholds (e.g., γ, δ) without extensive sensitivity studies; performance/feasibility trade-offs could vary with these choices.
5. **Clarity around “guaranteeing feasibility” vs “improving quality” could be sharper:** Since several baselines enforce feasibility via pruning or refusal, the paper should more explicitly attribute where feasibility guarantees come from, and focus the comparative advantage on budget efficiency, success rate, and overhead.

---

> ### Author Rebuttal · Authors · 2026-03-30
>
> We sincerely thank you for your time and valuable feedback during the review process! Our response is organized into two parts: (1) an index of supplementary experimental results addressing the concerns raised by all reviewers, and (2) point-by-point responses to each of your specific concerns.
>
> ---
>
> **[Section 1]  Supplementary Experiments**
>
> 1. Calibration curve of intention predictor [[Link](https://anonymous.4open.science/r/icml_agent-A77F/1.png)]
> 2. Fine-grained component ablation [[Link](https://anonymous.4open.science/r/icml_agent-A77F/2.png)]
> 3. Impact of additional mechanism [[Link](https://anonymous.4open.science/r/icml_agent-A77F/3.png)]
> 4. Sensitivity analysis [[Link](https://anonymous.4open.science/r/icml_agent-A77F/4.png)]
> 5. Open-source agent backbones [[Link](https://anonymous.4open.science/r/icml_agent-A77F/5.png)]
> 6. World model choice [[Link](https://anonymous.4open.science/r/icml_agent-A77F/6.png)]
> 7. Justification of LLM-as-a-Judge [[Link](https://anonymous.4open.science/r/icml_agent-A77F/7.png)]
> 8. Tool price distribution [[Link](https://anonymous.4open.science/r/icml_agent-A77F/8.png)]
>
> ---
>
> **[Section 2]  Responses**
>
> * **[W1, Q1] Pricing.**
>   We additionally evaluate INTENT under a **LogNormal** price distribution in **Exp. 8**, and the conclusions remain consistent. More importantly, our goal in the main setup is to construct a **tightly budget-constrained environment** that stresses whether the agent can make sound cost-sensitive decisions. Our price sampling is performed **per query**, meaning that the same tool may have very different prices across tasks. Under this design, uniform sampling gives every tool equal chance of becoming the cost bottleneck in some task, allowing us to more comprehensively test budget awareness. This differs fundamentally from prior setups such as **BTP** and **BATS**, where each tool is assigned a fixed price across tasks. In this sense, our setup is closer to a worst-case-style evaluation. We also note that varying the budget directly leads to consistent conclusions, as shown in **Fig. 5**.
>
> * **[W2, Q2] Calibration.**
>   We have added the calibration curve of the intention predictor in **Exp. 1**. The predictor is nearly perfectly calibrated on the test set (ECE = 7.7e-3) and remains consistent across subsets. When new tools emerge, the predictor becomes immediately ill-calibrated and tends to be overly optimistic on high-success-probability tools, which leads to a noticeable performance drop. As more interaction data is accumulated, however, periodic updates of the predictor yield progressively better ECE, and the agent’s performance shows a clear log-linear scaling trend (**Fig. 3**).
>
> * **[Q3, Q5] Hyperparameter sensitivity and ablation.**
>   We have added a sensitivity analysis for risk preference in **Exp. 4**. INTENT consistently outperforms the baselines across a broad range of values, while $\gamma$ provides an effective performance–cost trade-off. We also added fine-grained ablations of components (**Exp. 2**) and additional mechanisms (**Exp. 3**), clearly showing the contribution of major components. To make the method behavior more intuitive to readers, we will also include a **case study** in the final version (**ID: 70610**), which helps illustrate representative success and failure modes.
>
> * **[W3, Q4] Reliability of LLM-as-a-Judge.**
>   We additionally evaluate with judges from two other model families (**Gemini** and **Grok**) as well as **human experts** in **Exp. 7**. The ranking of methods remains consistent with that reported in the paper. We also note that ToolBench itself reports an agreement rate of **87.1%** between its PR metric and human expert evaluation [[Link](https://github.com/OpenBMB/ToolBench)], which further supports the validity of this protocol.
>
> * **[W5]**
>   Thank you for this helpful suggestion. We agree that the paper should more clearly distinguish feasibility from performance/cost efficiency. To satisfy a hard budget constraint, purely passive blocking of over-budget tool calls (e.g., DFSDT) is feasible but often overly aggressive. Methods that attempt to actively alter tool choices either rely on heuristics (BATS) or do not account for the agent’s dynamic decision process (BTP). In contrast, INTENT provides a principled, dynamic, and quantifiable planning mechanism. Empirically, under the same hard constraint, INTENT achieves **near-upper-bound performance** (**Fig. 1**) with **the lowest cost** and only **moderate overhead** (**Tab. 1**). We will revise the presentation in the final version to make this comparative advantage—namely, better **performance–cost efficiency** under hard constraints—more explicit.
>
> ---
>
> We sincerely thank you again for your time and effort during the review process! We hope our responses have addressed your concerns. If there are any remaining issues or further questions, please feel free to let us know—we would be more than happy to continue the discussion!

---

> > ### Author Rebuttal · Reviewer_Wy1B · 2026-04-01
> >
> > Thank you for the thoughtful rebuttal. The response has addressed my main concerns in a clear and constructive way, and it has increased my confidence in the paper. I continue to view the paper as a technically solid and meaningful contribution on an important problem, and I maintain my positive overall assessment. I am keeping my Overall Recommendation unchanged at 4 (Weak Accept), but I am increasing my Confidence score from 3 to 4.

---

### Official Review · Reviewer_cPfz · 2026-03-05

**Soundness:** 2
**Presentation:** 3
**Significance:** 3
**Originality:** 2
**Overall Recommendation:** 3
**Confidence:** 4

**Summary:**

This paper studies the problem of budget-constrained tool use in agentic large language models. In this setting, an LLM must solve multi-step tasks using external tools while respecting a strict monetary budget, where each tool invocation incurs a cost.

To address this challenge, the paper proposes INTENT, an inference-time planning framework that augments a pretrained LLM agent with a learned language world model and an intention-based cost estimation mechanism. Instead of performing expensive search such as Monte Carlo Tree Search, the framework performs lightweight single-trajectory simulations using a world model to anticipate future tool usage.

A key idea of the method is to decompose tool outcomes into two components: whether the tool result satisfies the agent’s semantic intention and the specific content of the tool output. The system estimates the probability of intention satisfaction and derives a pessimistic expected cost estimate based on this probability, allowing the framework to decide whether a proposed tool action should be executed under the remaining budget.

Experiments on cost-augmented StableToolBench demonstrate improvements in pass rate and budget feasibility compared to several baseline approaches under both non-reasoning and reasoning model backbones.

**Compliance With Llm Reviewing Policy:**

Affirmed.

**Final Justification:**

The paper addresses a practically relevant problem of budget-constrained tool use for LLM agents, with a coherent system design integrating an intention predictor and planning logic. However, upon reflection and in light of the AC's comment regarding the role of post-submission results in the acceptance decision, I believe my evaluation should primarily rest on the originally submitted evidence.

In the original submission, two of my core concerns had no supporting evidence:

- **Generality**: all experiments were conducted exclusively on GPT-family models
- **Evaluation validity**: LLM-as-a-Judge relied solely on same-family models without cross-family or human validation

Additionally, two further concerns remain unresolved even after the rebuttal:

- **System-level cost analysis**: still qualitative rather than quantitative
- **Single benchmark**: evaluation relies solely on StableToolBench

I also want to note that the paper's framing appears broader than its actual contribution. The title invokes terms such as "Agentic AI" and "World Models," yet the evaluation relies solely on StableToolBench with synthetic prices, which is insufficient to substantiate claims of such generality. A more measured framing that accurately reflects the scope of the contribution would strengthen the paper.

I am revising my score from 4 (Weak Accept) back to 3 (Weak Reject) to reflect an assessment grounded in the original submission.

**Key Questions For Authors:**

1. The experiments are conducted exclusively with GPT-family models (GPT-4.1 mini and GPT-5 nano) as the agent backbone. It would be helpful to understand whether the proposed approach generalizes to other model families.
Could the authors provide results or discussion on applying INTENT to open-source models such as LLaMA, Qwen, or other widely used open LLMs?

2. The evaluation relies on an LLM-as-a-Judge setup, where both the evaluated agents and the judge model belong to the GPT family. This raises the possibility of familiarity bias in the evaluation.
Could the authors clarify how they ensure that the results are not affected by such bias?

3. A key motivation of the work is budget-aware tool usage. However, the proposed method introduces additional components, including a language world model and an intention predictor, and performs simulation rollouts during inference.
Could the authors provide a more comprehensive system-level cost analysis that includes both tool execution costs and the additional inference costs introduced by the planning framework?

4. The world model and conditional generator are implemented using Qwen2.5-3B-Instruct. It would be useful to better understand how sensitive the method is to this design choice.
Could the authors provide additional analysis on the impact of the world model architecture and scale? For example, would smaller models or alternative architectures lead to similar performance?

**Limitations:**

yes

**Strengths And Weaknesses:**

Strengths:

- The paper studies a practically relevant problem: budget-aware tool usage for LLM agents interacting with external APIs.

- The formulation of budget-constrained agentic tool use as a sequential decision-making problem with explicit monetary costs is interesting and timely given the increasing deployment of tool-based agents.

- The overall system design is logically structured and clearly described, with a coherent integration of the world model, intention predictor, and planning logic.


Weaknesses:

- The experimental evaluation relies entirely on proprietary models (GPT-4.1 mini and GPT-5 nano) as the underlying agents. No open-source models are evaluated. This makes it difficult to assess the generality of the proposed approach and raises concerns about reproducibility.

- The benchmark used in the evaluation (StableToolBench) relies on LLM-as-a-Judge evaluation. In this work, the judge model and the evaluated agent models belong to the same model family (GPT). This creates a potential familiarity bias, where the evaluation model may systematically prefer outputs produced by models with similar reasoning style or training distribution.

- The central motivation of the paper is budget efficiency, yet the proposed framework introduces additional learned models (a 3B language world model and an additional intention predictor) and performs simulation rollouts during inference. The paper does not provide a detailed end-to-end cost analysis comparing tool savings versus additional LLM inference costs, making it unclear whether the method truly reduces overall system cost.

- The evaluation is limited to a single benchmark setting. Additional experiments on other tool-use or agent benchmarks would help demonstrate the robustness and general applicability of the method.

---

> ### Author Rebuttal · Authors · 2026-03-30
>
> We sincerely thank you for your time and valuable feedback during the review process! Our response is organized into two parts: (1) an index of supplementary experimental results addressing the concerns raised by all reviewers, and (2) point-by-point responses to each of your specific concerns.
>
> ---
>
> **[Sec. 1]**
>
> 1. Calibration curve of intention predictor [[Link](https://anonymous.4open.science/r/icml_agent-A77F/1.png)]
> 2. Fine-grained component ablation [[Link](https://anonymous.4open.science/r/icml_agent-A77F/2.png)]
> 3. Impact of additional mechanism [[Link](https://anonymous.4open.science/r/icml_agent-A77F/3.png)]
> 4. Sensitivity analysis [[Link](https://anonymous.4open.science/r/icml_agent-A77F/4.png)]
> 5. Open-source agent backbones [[Link](https://anonymous.4open.science/r/icml_agent-A77F/5.png)]
> 6. World model choice [[Link](https://anonymous.4open.science/r/icml_agent-A77F/6.png)]
> 7. Justification of LLM-as-a-Judge [[Link](https://anonymous.4open.science/r/icml_agent-A77F/7.png)]
> 8. Tool price distribution [[Link](https://anonymous.4open.science/r/icml_agent-A77F/8.png)]
>
> ---
>
> **[Sec. 2]**
>
> * **[W1, Q1] Generality.**
>   We have added results on two representative open-source agent backbones, **Qwen3.5-35B-A3B** and **DeepSeek-V3.2**, in **Exp. 5**. The conclusions are consistent with those on the closed-source backbones: INTENT continues to deliver clear gains while maintaining strict budget feasibility.
>
> * **[W2, Q2] Potential evaluator bias.**
>   We additionally evaluate with judges from two other model families (**Gemini** and **Grok**) as well as **human experts** in **Exp. 7**. The ranking of methods remains consistent with that reported in the paper. We also note that ToolBench itself reports an agreement rate of 87.1% between its PR metric and human expert evaluation [[Link](https://github.com/OpenBMB/ToolBench)]. More importantly, even if some judge bias exists, all compared inference-time methods in our experiments are built on the **same agent backbone** within each setting, so the relative comparisons remain fair.
>
> * **[Q4] Sensitivity to world model choice.**
>   We have added experiments with world models of different scales (**0.5B–7B**) and architectures (**Llama** and **Qwen**) in **Exp. 6**. The results are overall stable across these choices. In particular, a relatively small frontier open-source model is already sufficient to achieve near-best performance.
>
> * We also took your concern on soundness seriously and added several supplementary analyses, including fine-grained ablations of components and additional mechanisms (**Exp. 2, 3**),  and sensitivity analysis for risk preference (**Exp. 4**), further supporting our results.
>
> * **[W3, Q3] System-level cost analysis.**
>   Our intended deployment scenario is that a model provider leverages historical tool-interaction logs to train a **lightweight oracle** that equips a strong base agent with budget-aware behavior. At inference time, the additional overhead comes from two sources:
>   (i) inference of the **world model** and **intention predictor**, and
>   (ii) extra **agent tokens** introduced by simulation.
>   The first source is minor, since these auxiliary models (3B and 0.6B) are orders of magnitude smaller than the underlying agent, and their workloads are substantially lighter than long-chain reasoning by the agent itself. For the second source, we already report the token overhead in the main paper (**Tab. 1**), and the increase is moderate. While tool cost and token cost are measured in different units and therefore cannot be directly added, we believe the practical value of our approach is clear: it uses cheap computation to reduce costly external tool calls. Given the drastic decline in inference cost for a given level of intelligence, we believe this trade-off is increasingly realistic in practice.
>
> * **[W4]** Thank you for raising this point. We also explored a broad range of existing agentic and tool-use benchmarks during the project. However, many of them are either extremely expensive to evaluate and difficult to retrofit with explicit tool pricing (e.g., MCP-Universe, LiveMCP-101), involve only a few tools (e.g., WebWalker, GAIA, xbench), or ensure unique solution path (e.g., BFCL). To the best of our knowledge, ToolBench is currently the only suitable benchmark for our setting: it contains a large number of real APIs (16k), supports diverse solution paths, includes a substantial number of test instances (765) spanning 49 categories, and has been widely adopted both by prior work (e.g., BTP) and by the community more broadly. If more realistic benchmarks become available, we would be very happy to evaluate INTENT on them as well.
>
> ---
>
> We sincerely thank you again for your time and effort during the review process! We hope our responses have addressed your concerns. If there are any remaining concerns that still leave you uncertain about this work, please let us know—we would be more than happy to continue the discussion!

---

> > ### Author Rebuttal · Reviewer_cPfz · 2026-04-03
> >
> > Thank you for the thorough rebuttal. The response addresses my main concerns clearly and constructively. The additional experiments on open-source backbones (Exp. 5), world model architecture/scale sensitivity (Exp. 6), and cross-family judge evaluation (Exp. 7) are encouraging and substantially strengthen the paper. However, the system-level cost analysis remains qualitative rather than quantitative, and the reliance on a single benchmark still limits confidence in the generality of the findings. After clarification, the paper seems technically sound and meaningful if the authors incorporate the additional results into the revised version. Therefore, I am raising my score. I have changed my overall recommendation to 4 (Weak Accept).

---

### Official Review · Reviewer_JubG · 2026-03-14

**Soundness:** 3
**Presentation:** 3
**Significance:** 3
**Originality:** 3
**Overall Recommendation:** 4
**Confidence:** 3

**Summary:**

This paper formalizes budget-constrained tool use for LLM agents as sequential decision making in context space with priced, stochastic tool calls. The authors propose INTENT, an inference-time planning framework that introduces an intention-based decomposition of tool outcomes: a binary latent variable z_t indicates whether a tool call satisfies the agent's semantic intention, enabling factorization into an Intention Predictor (estimating success probability ρ_t) and a Conditional Generator (producing ideal observations with z=1). INTENT simulates an "ideal trajectory" by forcing z=1 at every step to extract the agent's latent plan, then applies geometric cost calibration (c_hat_k = COST(a_k)/ρ_k) with a risk factor γ for budget-aware accept/reject decisions. On cost-augmented StableToolBench with two backbones (GPT 4.1 mini, GPT 5 nano), INTENT achieves 100% budget feasibility with substantial pass rate gains (63.8%/76.0%) over enforce baselines, and demonstrates robustness to price perturbations, new tools, and budget scaling.

**Compliance With Llm Reviewing Policy:**

Affirmed.

**Key Questions For Authors:**

1. How sensitive is INTENT to the risk factor γ and blacklist threshold δ? A sensitivity plot across a reasonable range would clarify whether the fixed values generalize or require task-specific tuning.

2. Could you provide component ablations — specifically, INTENT without the conditional generator (using the unconditional world model instead), and with a simpler cost model (e.g., fixed multiplier instead of geometric calibration) — to isolate where the gains originate?

3. The last-call cache accepts a previously rejected action without lookahead if re-proposed. How often does this bypass trigger, and does it lead to budget violations or suboptimal outcomes in tight-budget scenarios?

**Limitations:**

Partially. The paper acknowledges the inference-time overhead and the requirement for a trained world model, but does not discuss the sensitivity to synthetic pricing assumptions or the potential bias from the geometric retry model. The reliance on LLM-as-a-Judge for both training labels and evaluation deserves more discussion.

**Strengths And Weaknesses:**

### Strengths

- **Timely and practical problem formulation.** Budget-constrained tool use is a genuine deployment concern as agents interact with monetized APIs. The inference-time approach that does not modify the base agent is directly deployable. The formalization as sequential decision making with hard budget constraints is clean.

- **Effective intention-based decomposition.** Separating whether a tool call succeeds (z_t) from what it returns is a principled way to handle output stochasticity. The ideal trajectory simulation elegantly extracts the agent's latent plan, and geometric cost calibration converts success probabilities into risk-adjusted budgets. This is a meaningful conceptual contribution.

- **Strong empirical results with robustness studies.** INTENT achieves the best performance-feasibility trade-off across both backbones. The robustness experiments (new tool exposure via data subsampling, price perturbations, budget scaling) are well-designed and show stable performance under market non-stationarity, with log-linear scaling in the low-data regime being particularly compelling.

### Weaknesses

- **No ablation on key hyperparameters.** γ=0.5 and δ=0.1 are fixed across all experiments without sensitivity analysis. The paper also lacks component ablations (e.g., INTENT without conditional generator, or replacing geometric calibration with simpler heuristics). It is unclear how much each component contributes to the overall gains and how sensitive performance is to these choices.

- **Synthetic price model limits external validity.** All tool prices are sampled from U(5,50) with a fixed budget B=50. Real-world API pricing involves tiered structures, fixed fees, and heavy-tailed distributions. The absence of experiments with more realistic price models or actual API costs weakens the claim of practical applicability.

- **Geometric retry model is restrictive.** The cost calibration assumes a constant success probability across retries (geometric distribution), ignoring that agents adapt arguments and strategies between attempts. This can systematically bias cost estimates — overestimating for adaptive agents, underestimating when tools have correlated failures.

---

> ### Author Rebuttal · Authors · 2026-03-30
>
> We sincerely thank you for your time and valuable feedback during the review process! Our response is organized into two parts: (1) an index of supplementary experimental results addressing the concerns raised by all reviewers, and (2) point-by-point responses to each of your specific concerns.
>
> ---
>
> **[Section 1] Supplementary Experiments**
>
> 1. Calibration curve of intention predictor [[Link](https://anonymous.4open.science/r/icml_agent-A77F/1.png)]
> 2. Fine-grained component ablation [[Link](https://anonymous.4open.science/r/icml_agent-A77F/2.png)]
> 3. Impact of additional mechanism [[Link](https://anonymous.4open.science/r/icml_agent-A77F/3.png)]
> 4. Sensitivity analysis [[Link](https://anonymous.4open.science/r/icml_agent-A77F/4.png)]
> 5. Open-source agent backbones [[Link](https://anonymous.4open.science/r/icml_agent-A77F/5.png)]
> 6. World model choice [[Link](https://anonymous.4open.science/r/icml_agent-A77F/6.png)]
> 7. Justification of LLM-as-a-Judge [[Link](https://anonymous.4open.science/r/icml_agent-A77F/7.png)]
> 8. Tool price distribution [[Link](https://anonymous.4open.science/r/icml_agent-A77F/8.png)]
>
> ---
>
> **[Section 2] Response**
> - **[W1, Q1] Sensitivity to $\gamma$.**
>     We have added a sensitivity analysis for risk preference parameter $\gamma$ in **Exp. 4**. INTENT consistently outperforms the baselines across a broad range of values, while $\gamma$ provides an effective performance–cost trade-off.
> - **[W1, Q2] Component ablations.**
>     We have added fine-grained component ablations in **Exp. 2**. The results show that replacing any major component with a simpler alternative leads to a clear performance drop, supporting the contribution of each module. We further provide ablations of the additional mechanisms, including the blacklist ($\delta$), in **Exp. 3**. These results suggest that INTENT’s gains do not rely on such engineering additions; they mainly improve efficiency rather than core effectiveness.
> - **[W2] Synthetic pricing.**
>     We agree that real-world API pricing can be much more complex. To address this concern, we additionally evaluate INTENT under a **LogNormal** price distribution in **Exp. 8**, and the conclusions remain consistent. More importantly, our goal in the main setup is to construct a **tightly budget-constrained environment** that stresses whether the agent can make sound cost-sensitive decisions. Our price sampling is performed **per query**, meaning that the same tool may have very different prices across tasks. Under this design, uniform sampling gives every tool a comparable chance of becoming the cost bottleneck in some task, allowing us to more comprehensively test budget awareness. This differs fundamentally from prior setups such as **BTP** and **BATS**, where each tool is assigned a fixed price across tasks. In this sense, our setup is closer to a stress test or worst-case-style evaluation. We also note that varying the budget directly leads to consistent conclusions, as shown in **Fig. 5**.
> - **[W3] Geometric retry model.**
>     We agree that the geometric retry assumption is idealized, and a more refined treatment of systematic bias under adaptive retries is an important direction for future work. That said, this assumption plays a key practical role in INTENT: it allows us to reduce lookahead to a **single simulation**, which is essential for keeping inference tractable. Empirically, the good calibration of the intention predictor (**Exp. 1**) and the ablation results (**Exp. 2**) both support the effectiveness of this approximation in practice. We therefore view it as a useful and principled approximation that enables lightweight budget-aware planning, while acknowledging that richer retry models are worth exploring in future work.
> - **[Q3] Last-call cache.**
>     The last-call cache still enforces the **immediate budget check** (**Alg. 3, line 4**). Therefore, even if a previously rejected tool call is re-proposed, it will not be executed if the call itself would already exceed the remaining budget. As shown in **Exp. 3**, this mechanism mainly improves efficiency by avoiding repeated lookahead simulations in cases where the same tool appears necessary from the agent’s perspective. Its effect on final performance is limited, and we do not observe it causing budget violations.
> ---
>
> We sincerely thank you again for your time and effort during the review process! We hope our responses have addressed your concerns. If there are any remaining issues or further questions, please feel free to let us know—we would be more than happy to continue the discussion!

---

### Decision · Program_Chairs · 2026-04-30

**Decision:**

Reject

**Comment:**

The paper addresses an important problem and proposes a technically interesting approach, but the post-rebuttal discussion made clear that the case for acceptance is not strong when judged on the original submission. In particular, the reviewers agreed that substantial new experiments introduced during rebuttal should not materially change the basis for the decision. That discussion led to a weaker overall assessment than the initial scores alone would suggest.

The main issue is that the original submission did not provide sufficiently strong empirical support for its claims. The evaluation was confined to a single benchmark, depended heavily on GPT-family models and same-family LLM-as-a-judge evaluation, and did not include a convincing quantitative system-level cost analysis despite the central emphasis on budget efficiency. These are not minor gaps. They directly limit confidence in the generality, validity, and practical significance of the results.

The rebuttal did provide extensive additional experiments, but these went far beyond modest clarification and are difficult to evaluate with the same confidence as results that were part of the submitted paper. Even with those added results, concerns remained about the breadth of validation and the paper’s overall scope. On balance, the original submission did not make a strong enough case for acceptance.